# Structural and Neuronal Integrity Measures of Fatigue Severity in Multiple Sclerosis

**DOI:** 10.3390/brainsci7080102

**Published:** 2017-08-12

**Authors:** Evanthia Bernitsas, Kalyan Yarraguntla, Fen Bao, Rishi Sood, Carla Santiago-Martinez, Rajkumar Govindan, Omar Khan, Navid Seraji-Bozorgzad

**Affiliations:** 1Multiple Sclerosis Center, Department of Neurology, Detroit Medical Center, 4201 St Antoine, 8C-UHC Detroit 48201, MI, USA; rsood@med.wayne.edu (R.S.); du2067@wayne.edu (C.S.-M.); okhan@med.wayne.edu (O.K.); 2The Sastry Foundation Advanced Imaging Laboratory, Wayne State School of Medicine, 4201 St Antoine, Detroit 48201, MI, USA; kalyancy@wayne.edu (K.Y.); fbao@med.wayne.edu (F.B.); nseraji@wayne.edu (N.S.-B.); 3Department of Pediatric Neurology/PET Center, Children’s Hospital of Michigan, 4201 St Antoine, Detroit 48201, MI, USA; mgrajkumar45@gmail.com

**Keywords:** fatigue, multiple sclerosis, diffuse tensor imaging, fatigue severity scale, deep gray matter nuclei volume, cortical thickness

## Abstract

Fatigue is a common and disabling symptom in Multiple Sclerosis (MS). However, consistent neuroimaging correlates of its severity are not fully elucidated. In this article, we study the neuronal correlates of fatigue severity in MS. Forty-three Relapsing Remitting MS (RRMS) patients with MS-related fatigue (Fatigue Severity Scale (FSS) range: 1–7) and Expanded Disability Status Scale (EDSS) ≤ 4, were divided into high fatigue (HF, FSS ≥ 5.1) and low fatigue groups (LF, FSS ≤ 3). We measured T2 lesion load using a semi-automated technique. Cortical thickness, volume of sub-cortical nuclei, and brainstem structures were measured using Freesurfer. Cortical Diffusion Tensor Imaging (DTI) parameters were extracted using a cross modality technique. A correlation analysis was performed between FSS, volumetric, and DTI indices across all patients. HF patients showed significantly lower volume of thalamus, (*p* = 0.02), pallidum (*p* = 0.01), and superior cerebellar peduncle ((SCP), *p =* 0.002). The inverse correlation between the FSS score and the above volumes was significant in the total study population. In the right temporal cortex (RTC), the Radial Diffusivity ((RD), *p =* 0.01) and Fractional Anisotropy ((FA), *p =* 0.01) was significantly higher and lower, respectively, in the HF group. After Bonferroni correction, thalamic volume, FA-RTC, and RD-RTC remained statistically significant. Multivariate regression analysis identified FA-RTC as the best predictor of fatigue severity. Our data suggest an association between fatigue severity and volumetric changes of thalamus, pallidum, and SCP. Early neuronal injury in the RTC is implicated in the pathogenesis of MS-related fatigue.

## 1. Introduction

Multiple Sclerosis (MS) is a chronic immune-mediated central nervous system disease and a leading cause of non-traumatic disability in the young adult population [1]. Up to 80% of patients with MS report fatigue that severely impacts their daily activities, quality of life, and employment status, frequently leading to part-time employment or early retirement [2,3,4]. Furthermore, the impairment resulting from MS-related fatigue is recognized by the United States Social Security Administration as a criterion for disability [2,3]. 

MS-related fatigue is characterized by the constant feeling of exhaustion and limited endurance of sustained physical and mental activities [4,5,6]. Since fatigue is a subjective symptom with a physical and mental component, the objective assessment of its severity is often challenging. The Fatigue Severity Scale (FSS) and Modified Fatigue Impact Scale (MFIS) are the most commonly used measures of fatigue severity. FSS is reported to have higher test-retest consistency compared to MFIS [7,8,9]. It seems that MS-related fatigue is multidimensional and cannot merely be explained by the degree of clinical disease activity, neurological disability, or the extent of Magnetic Resonance Imaging (MRI) abnormalities [10]. Although the pathology of MS-related fatigue is not clear, previous studies have reported atrophy of gray (GM) and white matter (WM), disruptions in cortico-subcortical connections involving the fronto-parietal cortex, and reduction in thalamus and basal ganglia nuclei volume in MS patients compared to healthy controls [11,12,13]. In contrast, other studies reported no correlation between fatigue severity and white matter disease, and the role of cortical atrophy in MS-related fatigue remains controversial [14,15]. In this retrospective cross-sectional study, we used volumetric and diffusion metrics to investigate the anatomical and neuronal integrity of specific brain structures in relation to fatigue severity in MS patients with low disability. 

## 2. Methods

### 2.1. Participant Recruitment and Selection Criteria

Forty-three relapsing remitting multiple sclerosis patients (RRMS) patients were enrolled in this retrospective cross-sectional study from the MS Center, Division of Neurology, Detroit Medical Center, Michigan. We included patients from 18 to 65 years of age, diagnosed with MS per the revised McDonald criteria. Patients had a relapsing remitting course and were relapse and steroid treatment-free for at least one month prior to MRI scan. We excluded patients who were pregnant or had other neurological or psychiatric disorders, such as depression or anxiety, because of their established association with fatigue [16]. Patient on antidepressants, psychoactive medications, stimulants, or medications for symptomatic treatment of fatigue were excluded. All included patients denied sleep disorders and other causes of fatigue such as active infection, malignancy, anemia, thyroid or adrenal disease. On the same day of MRI acquisition, MS patients underwent a neurological evaluation, including the Expanded Disability Status Scale (EDSS). Patients with an EDSS > 4 were excluded in order to minimize the effect of physical disability on fatigue [17]. Fatigue severity was assessed using FSS, given the higher Cronbach’s alpha value (0.89) of FSS compared to MFIS (0.81) [7,8]. FSS is a self-report questionnaire consisting of nine statements with a seven-point scale response per statement, with lower scores indicating less fatigue [18]. The RRMS patients with a mean FSS score ≥ 5.1 were categorized as high fatigued (HF), those with a mean FSS score ≤ 3 as low fatigued (LF), and those between FSS score 3.1–5 were classified as moderately fatigued (MF). The effect of various MS medications was minimal, as all the patients included in this study were on fingolimod.

The study was approved by the Wayne State University Institutional Review Board. A signed informed consent was obtained from all enrolled participants.

### 2.2. MRI Image Acquisition 

Whole-brain MRI scan was performed using a 3-Tesla Siemens Verio System (Siemens Medical Systems, Erlangen, Germany). The following protocols were used for this analysis: (1) localizer sequence, (2) 3-D T1 weighted Magnetization Prepared Rapid Acquisition Gradient Echo (MPRAGE) images [Repetition time/Echo time (TR/TE) = 1680/3.52 ms, flip angle 9°, acquisition matrix size 384 × 384, with 176 slices, giving a nominal voxel size of 0.7 × 0.7 × 1.3 mm], and (3) a DTI sequence using a single-shot spin-echo diffusion sensitized echo-planar imaging sequence with balanced Icosa21 tensor encoding scheme (TR/TE = 10,400/126 ms, flip angle 90°, acquisition matrix size 200 × 200, with 46 consecutive slices, giving a voxel size of 1.3 × 1.3 × 3 mm).

### 2.3. MRI Data Processing and Analysis

Cortical reconstruction and volumetric segmentation was obtained from the 3-D T1 images using Freesurfer image analysis version 5 [19] as described in prior publications [20,21,22,23]. We used the standard protocol including skull stripping and image registration to Talairach brain atlas, followed by segmentation, topology correction, and placement of gray matter-white matter (GM-WM), white matter- cerebrospinal fluid (WM-CSF) boundaries using surface normalization and intensity gradients. In addition to the cortical thickness and volume of subcortical structures that were obtained from Freesurfer, we used the coordinates of the cortical boundaries to extract the diffusion parameters from the DTI images, as follows. 

The diffusivity maps—no diffusion (b0), Mean Diffusivity (ADC), Radial Diffusivity (RD) and Fractional Anisotropy (FA) maps—were generated from the DTI sequence using DTI studio 3.0 [24]. We used FSL software to spatially register the Freesurfer brain-extracted FA images of each subject onto the subject’s 3-D T1 images using a simple rigid body (six degrees of freedom) registration, followed by a rigorous non-linear image registration (FNIRT) [23]. FA, ADC, and RD images were then resampled to the structural image space for further analysis. Subsequently, the reconstructed Gray-White and the Gray-Pial interface surfaces of the structural image from Freesurfer analysis were used as the boundaries of the gray and the white matter and to generate new surface images [25], which were then used to obtain the average diffusion value along the normal vector. Finally, the labeled masks of frontal, temporal, parietal, and occipital lobes were used to measure the FA, ADC, and RD values for each subject. 

### 2.4. Statistical Methods

Population demographics, diffusivity parameters of cortices, and subcortical nuclei volume variation between HF and LF groups were analyzed using an independent *t*-test. Boot strapping was performed to avoid the assumption of normality. A Mann-Whitney test was used to verify the findings. We did not find any discrepancy between the results from the non-parametric and boot-strapped *t*-tests. We used the Bonferroni method to correct for multiple comparisons, given the large number of variables used to model the fatigue scores, and both uncorrected and corrected *p*-values are reported. Pearson correlation with bootstrapping was used to explore the relation between the fatigue score and volumetric or DTI measures. A binomial, multivariate regression analysis was used to evaluate the best predictors of fatigue score from among those independent variables that showed a significant difference between the high and low fatigue scores. Two-sided bootstrapped *p*-value < 0.05 was considered statistically significant. All results are expressed as means ± standard error of mean (SEM), and statistical analysis was performed using SPSS v23.(International Business Machines Corporation, SPSS statistics for Windows, version 23.0, Armonk, NY, United States)

## 3. Results

### 3.1. Demographics and Clinical Data

Overall, 43 patients with RRMS participated: 15 patients were classified as the HF group, 14 as the MF, and 14 as the LF group. The groups did not differ in age, disease duration, medication, EDSS, or T2 lesion volume. The sample was primarily female (*n* = 26) with a mean age of 41 (±2.4) years. Seventeen patients were men with a mean age of 39 (±2.3) years. Demographic and clinical characteristics are shown in Table 1. 

### 3.2. Structural Imaging Findings

Given the number of variables used in this study, we first looked at those variables that were most strikingly different between the HF and LF groups. The subcortical nuclei volumes were lower in the high fatigue group compared to the low fatigue group, prior to correcting for multiple comparisons: thalamic (HF: 11.5 ± 0.3 mL vs. LF: 14 ± 0.6 mL; *p* = 0.001), pallidal (HF: 2.6 ± 0.07 mL vs. LF: 3 ± 0.13 mL; *p* = 0.013), and SCP (HF: 207.3 ± 7.1 mL vs. LF: 246.1 ± 9.6 mL; *p* = 002, Figure 1a–c). Of these structures, only the thalamic volume retained statistical significance after Bonferroni correction for multiple comparisons (corrected *p*-value = 0.007, Appendix A). In addition to the HF and LF group differences, we looked at the correlation between the fatigue score and the volume of thalamus, pallidi, and SCP across all patients. Figure 2a–c show the inverse correlation with the FSS score: thalamic (*r*^2^ = 0.416, *p* = 0.006), pallidal (*r*^2^ = 0.399, *p* = 0.005), and SCP (*r*^2^ = 0.293, *p* = 0.04). The relationship between these brain structure volumes and EDSS was not significant.

### 3.3. Diffuse Tensor Imaging Findings

A significant difference in diffusion tensor parameters was observed in the right temporal cortex (RTC) between the HF and LF groups. Radial Diffusivity (HF: 8.82 × 10^−4^ ± 0.2 × 10^−4^ mm^2^/s vs. LF: 8.18 × 10^−4^ ± 0.13 × 10^−4^ mm^2^/s; *p* = 0.016) was significantly higher in the HF group compared to the LF group (Figure 1d). In contrast, the HF group had significantly lower FA compared to the LF group (HF: 2.44 × 10^−1^ ± 0.04 × 10^−1^ vs. LF: 2.71 × 10^−1^ ± 0.08 × 10^−1^; *p* = 0.004) in RTC (Figure 1e). Both RD and FA retained statistical significance after correction for multiple comparisons (Bonferroni uncorrected *p*-values = 0.004 and 0.016, corrected *p*-values = 0.005 and 0.026 for FA and RD, respectively). Furthermore, the RD of RTC (*r*^2^ = 0.349, *p* = 0.01) had a significant inverse correlation with the FSS score across all patients (Figure 2d), and the FA of RTC (*r*^2^ = 0.358, *p* = 0.01) had a significant positive correlation with the FSS score (Figure 2e). The RD and FA of other cortices was not significantly different between the groups, and the variation in ADC of RTC did not reach statistical significance between the groups (Appendix A).

### 3.4. Multivariate Regression Analysis

We further explored the best MRI correlates of fatigue score in our patient group using a binomial regression analysis. The binomial defendant variable representing HF or LF was modeled using the independent variables that we found to be significantly different between the two groups in the univariate analysis, namely, subcortical volumes (thalamus, pallidum, and SCP), as well as FA and RD of the right temporal cortex. Since fatigue can be significantly affected by age, we included age in one model; however, since our sample did not have a large age difference between the groups, we also ran the model without age as a variable to avoid overestimation. Results are presented in Table 2, and demonstrate that in a multivariate regression model, right temporal cortex fractional anisotropy is the best correlate of fatigue status among the variables examined (*p* = 0.11 after correction for age, and *p* = 0.023 when age was excluded from the model).

## 4. Discussion

The main focus of our study was to explore the structural and neuronal integrity measures of fatigue severity in RRMS patients with low disability, using the combination of structural and diffuse tensor imaging techniques. The correlation between fatigue and disability status is debated. Flachenecker et al. showed that fatigue is strongly related to physical disability [7]. Biberacher et al. reported a significant correlation between fatigue severity and EDSS in all three study cohorts (discovery, MRI, and CSF validation cohorts) [26]. In contrast, Krupp et al. found no significant relationship between fatigue and EDSS score [27]. In a large prospective study, Bakshi et al. reported no significant association between fatigue severity and EDSS [28]. In our study, we recruited MS patients with low disability in an attempt to study a homogenous population, by minimizing the potential impact of disability on fatigue severity and potentially on volumetric and DTI indices. 

Given the complex nature of fatigue in MS, a wide variety of imaging techniques have been used in previous studies, and multi-regional damage rather than global brain damage was implicated. Research has identified a strong link between thalamic and basal ganglia nuclei abnormalities and the pathophysiology of fatigue [11,29,30,31]. A recent study reported lower thalamic volume of RRMS patients with fatigue compared to healthy controls [27,28]. Finke et al. described disrupted functional connectivity within the basal ganglia nuclei (including pallidi, putamen, and caudate nuclei), which correlates with fatigue severity in MS patients [29]. However, the literature on the anatomical variation of the aforementioned structures and their possible correlation with fatigue severity and lesion load in RRMS is limited. 

In our study, we initially divided the MS patients into low and high fatigue groups, and eliminated patients with moderate fatigue, in order to augment potential groups’ differences in cortical thicknesses, subcortical volumes, or diffusion indices. We observed significantly lower thalamic, pallidal, and superior cerebellar peduncles volumes in the HF group compared to the LF group. Furthermore, the magnitude of thalamic, pallidal, and SCP atrophy correlated positively with the FSS score in the total study population, which suggests that these structures are affected proportionately to fatigue severity in RRMS. Our observations support prior findings obtained by using different approaches, such as functional Magnetic Resonance Imaging (fMRI )and voxel based morphometry [29,32]. Nourbakhsh et al. [33], studied a cohort of early relapsing MS patients and reported an association between lower thalamic volume at baseline and increasing physical subscale of MFIS during the study. They also found a trend for baseline thalamic and cerebellar cortical volume to predict subsequent change in total MFIS in the same cohort of patients. 

Previous studies have reported the involvement of cortical white matter in the fatigue pathology of RRMS patients. Notably, disruption of the fronto-parietal pathways was implicated based on increased FA in their respective white matter tract [6,13]. Advanced imaging methods, such as fluorodeoxyglucose Positron Emission Tomography (PET) have demonstrated dysfunction of the basal ganglia, fronto-parietal pathways, and cerebellar vermis [30]. Given the wide variability in imaging techniques used to study the cortical pathology of MS-related fatigue, we implemented a unique cross modality technique of registering DTI images to structurally sound MPRAGE data and obtained the DTI measures of their respective cortices [25]. The higher RD in parallel to lower FA indicated more severe neuronal injury in the right temporal cortex (RTC) of the HF group compared to the LF group. Furthermore, the RD and FA having direct and inverse correlation, respectively, with FSS score in RTC across all patients, suggest that the neuronal injury in RTC is proportionate to fatigue severity. Of note, RD and FA are not significantly affected in the left temporal cortex (LTC), which indicates neuronal integrity in the LTC. Additionally, the cortical thickness variation in RTC and in other cortices was not statistically significant between the study groups. These findings suggest that disruption of the neuronal integrity may occur prior to the evidence of cortical atrophy in the RTC. Moreover, they support and further expand previous evidence regarding the role of the right temporal lobe in the pathophysiology of fatigue. 

Rocca et al. reported significant atrophy in the right inferior temporal gyrus in pure MS fatigue patients compared to a healthy control group, using voxel-based morphometry [31]. Hanken et al. described cortical thinning in the right middle temporal lobe in a subgroup of patients with both fatigue and depression [34]. Bisecco et al. reported decreased FA and increased RD in the left superior longitudinal fasciculus (SLF) in a fatigued MS group. However, correlation analysis showed a significant association between higher FSS scores and lower FA at the level of right SLF [35]. Future studies on arcuate fasciculus, a part of the SLF that links the temporal cortex with the frontal and parietal lobes, may confirm these findings and elucidate its role in fatigue pathology. Given the initial clinical presentation of RRMS in patients enrolled, the aforementioned findings indicate that the RTC is involved in early fatigue pathology and may occur prior to the involvement of other cortices in RRMS. Given that the majority of studies in MS-related fatigue focus on white matter pathology, in our study we provide evidence for the neuronal integrity measures in cortical areas of RRMS patients. Gray matter imaging studies in MS will eliminate the confounding effect of the variability of FA due to MS lesions, and can still be able to detect the disease burden on the cortex and deep nuclei. Longitudinal studies with advanced gray matter imaging techniques may elucidate the role of the RTC in fatigue and could explain the role of other cortices in the prognosis of MS-related fatigue.

Previous studies have examined the relationship between T2 lesion load and fatigue severity and yielded conflicting results [36,37]. In our study, no significant correlation was found between fatigue severity and T2 lesion load. Strengths of this study include a homogenous study population consisting of patients with RRMS and low EDSS, careful selection criteria, close timing of MRI and fatigue assessment, a blinded MRI reader to clinical characteristics, and the application of robust statistical analyses. Our study is not without limitations. First, the small sample size, mainly due to the strict exclusion criteria, and the cross-sectional design, which interferes with group comparison, limit the generalization of our findings. Second, the location of MS lesions and their effect on the integrity of structures located in proximity to them, was not taken into account. Third, this study provides limited information on the structures involved in the fatigue circuitry of moderately fatigued RRMS patients. Finally, the exclusion of patients who were on stimulants or other medications for symptomatic treatment of fatigue may lead to sample bias, as the enrolled patients may represent a subgroup with the worst fatigue, resistant to therapeutic modalities. However, the robust statistical methods implemented to analyze the brain structures strengthen the significance of our observations, and the correlation we observed is unlikely to be affected by these shortcomings. 

## 5. Conclusions

In conclusion, our study shows that FA-RTC is the best predictor of fatigue severity in RRMS patients. Additionally, variation in thalamic volume may serve as a biomarker of fatigue in RRMS. Our study provides further evidence of an association between pallidal and SCP volume variation and fatigue severity, however, further investigation is required to reveal their role in MS-related fatigue. Moreover, higher RD in addition to reduced FA in the right temporal cortex indicates the early involvement of RD in fatigue pathology compared to other cortices in RRMS. Longitudinal integrated cortical DTI and volumetric studies in a large sample size are needed to confirm our findings and help in determining the prognosis of fatigue and its response to different treatment modalities.

## Figures and Tables

**Figure 1 brainsci-07-00102-f001:**
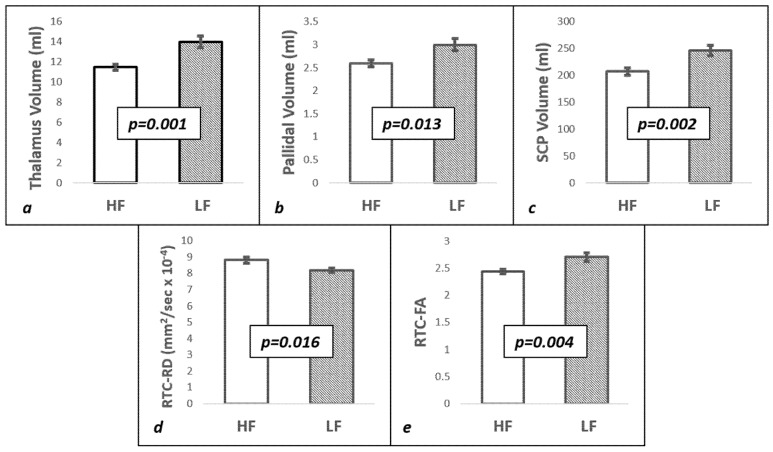
Volumetric and Diffusion measures showing lower Thalamic (**a**) Pallidal (**b**), Superior Cerebellar Peduncle (SCP, **c**) volume, and higher Radial Diffusivity (RD, **d**) and lower Fractional Anisotropy (FA, **e**) in Right Temporal Cortex (RTC) of High Fatigue group (HF) vs. Low Fatigue Group (LF). The error bars represent the standard error of mean.

**Figure 2 brainsci-07-00102-f002:**
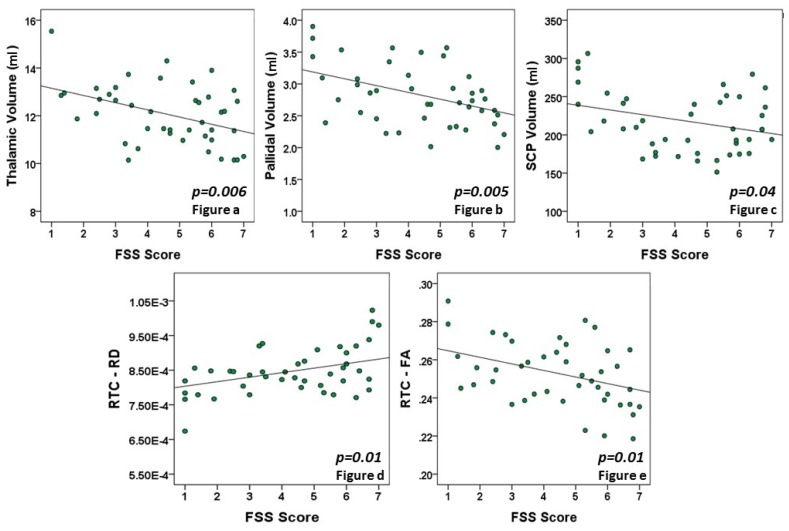
Correlation graphs showing inverse relation between Thalamic (**a**), Pallidal (**b**), Superior Cerebellar Peduncle (SCP, **c**) volume, Right Temporal Cortex-Fractional Anisotropy (RTC-FA, **e**) vs. FSS score, and positive relation between Right Temporal Cortex-Radial Diffusivity (RTC-RD, **d**) vs. Fatigue Severity Scale (FSS) score.

**Table 1 brainsci-07-00102-t001:** Demographic and clinical data.

RRMS Population	HF Group	MF Group	LF Group	Total	*p*-Value
Number of patients	15	14	14	43	
Ethnicity (Cau vs. AA)	9 vs. 6	6 vs. 8	9 vs. 5	24 vs. 19	
Age (years)	43 ± 2.9	39 ± 3	39 ± 1.7	41 ± 1.7	0.102
Range	(23–55)	(26–45)	(29–47)	(23–55)	
Mean FSS score	6 ± 0.12	4 ± 0.14	1.89 ± 0.2	4.35 ± 0.26	
Range	(5.1–7)	(3.1–5)	(1–3)	(1–7)	
Median EDSS score	2	2	1.5	2	0.754
Range	(1–4)	(1–4)	(1–4)	(1–4)	
T2 lesion volume (mL)	14 ± 2.5	18.8 ± 4.8	15.3 ± 5.9	15.6 ± 2.3	0.859
Range	(7.4–27.16)	(2.6–40.5)	(1.8–39.7)	(1.8–40.5)	
Disease period (years)	10 ± 1.7	9.2 ± 1.2	8.6 ± 1.9	9.3 ± 1	0.136
Range	(0.5–19.17)	(0.67–14.4)	(0.25–15)	(0.25–19.17)	

RRMS—Relapsing Remitting Multiple Sclerosis, FSS—Fatigue Severity Scale, EDSS—Expanded Disability Status Scale, HF—High Fatigue, MF—Moderate Fatigue, LF—Low Fatigue, M—Male, F—Female, Cau—Caucasian, AA—African American, ml—milliliter. The data represents average and standard error of mean.

**Table 2 brainsci-07-00102-t002:** (**a**). Regression model to predict fatigue level based on neural structures of significance. (**b**). Regression model, corrected for age, to predict fatigue level based on neural structures of significance. FA—Fractional Anisotropy, RD—Radial Diffusivity, RTC—Right Temporal Cortex, SCP-—Superior Cerebellar Peduncle.

**(a)**			
	**Standarized Beta Coefficient**	**Standard Error**	***p*-Value**
Thalamic volume	−1.042	0.514	0.043
Pallidal volume	0.068	1.435	0.962
SCP volume	−0.014	0.013	0.291
FA-RTC	−82.839	36.316	0.023
RD-RTC	−8567.04	10.782.51	0.427
**(b)**			
	**Standarized Beta Coefficient**	**Standard Error**	***p*-Value**
Age	0.119	0.061	0.058
Thalamic volume	−0.96	0.505	0.050
Pallidal volume	−0.14	1.474	0.924
SCP volume	−0.027	0.016	0.099
FA-RTC	−113.826	44.699	0.011
RD-RTC	−11070.1	10884.88	0.309

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
