# Peer review of "Structural and Neuronal Integrity Measures of Fatigue Severity in Multiple Sclerosis"

_brainsci, 2017, doi:10.3390/brainsci7080102_

Round 1

Reviewer 1 Report

The authors examined correlations among a collection of neural-level variables (white matter integrity: FA, RD; cerebral volumes derived through Freesurfer).

43 RRMS patients with low disability (EDSS <4) participated. Patients on antidepressants and fatigue modulating drugs (?) were excluded. There are currently no drugs accepted to treat MS fatigue. This exclusion criterion could potentially cause sample bias, as a large proportion of MS patients take drugs that may be considered fatigue-modulating, such as stimulants, which may be treating other underlying causes (e.g., ADHD). Also, many MS patients take anti-depresasnts to help them sleep. It may be patients with the worst fatigue who are trying pharmacological treatments, none of which has yet been identified as effective. This may actually be a very valuable population of patients to include in fatigue studies.

Please identify the disease-modifying agent subjects are on. Table 1, please provide disease duration in years.

Results of regression models to predict fatigue level (low n=14, moderate n=14, high n=15) would be a nice addition, first entering demographics (age, disease duration, EDSS) and T2 lesion volume, and adding structural volumes and DTI indices to the model next.

My main concern is that (as the authors acknowledge) the sample size is small, multiple comparisons have not been controlled for, and correlational research is limited. The small groups of patients falling into categories of low, moderate, and high fatigue are nicely matched for disease duration, T2 lesion load, and age. However, given the multidimensionality of fatigue (as mentioned by the authors), we must wonder whether relationships between neural level variables and level of fatigue are spurious and/or mediated by a third variable that has not been captured. The sample includes only patients with low disability (EDSS < 4); this should be mentioned in the abstract and the introduction, as it is a strength of the study. This methodological choice should be described in the context of previous MS literature and describe how this may affect results.

Language should be tempered in the abstract and discussion to reflect the study limitations. For instance, first sentence of discussion “focus of our study was to establish the structural and neuronal integrity measures of fatigue severity…” With 43 people, nothing has been or will be established.

Discussion: “These findings suggest that disruption of the neuronal integrity occurs prior to the evidence of cortical atrophy in the RTC.” This statement, with implications of a temporal sequence for observed (cross sectional) neural-level patterns, is an over interpretation of the study data and should be removed.

The statement “We provide novel evidence about the neuronal integrity measures in non-lesional cortical areas of RRMS patients” was a surprise; no accounting for lesions in Freesurfer or DTI analysis was described in the methods section. Final sentence of discussion the words “striking correlation” may be an overstatement.

Author Response

We would like to thank the reviewer for his/her constructive comments. We have revised the article accordingly as detailed below.

The authors examined correlations among a collection of neural-level variables (white matter integrity: FA, RD; cerebral volumes derived through Freesurfer).

43 RRMS patients with low disability (EDSS <4) participated. Patients on antidepressants and fatigue modulating drugs (?) were excluded. There are currently no drugs accepted to treat MS fatigue. This exclusion criterion could potentially cause sample bias, as a large proportion of MS patients take drugs that may be considered fatigue-modulating, such as stimulants, which may be treating other underlying causes (e.g., ADHD). Also, many MS patients take anti-depresasnts to help them sleep. It may be patients with the worst fatigue who are trying pharmacological treatments, none of which has yet been identified as effective. This may actually be a very valuable population of patients to include in fatigue studies.

Thank you for this excellent point. The term "fatigue-modulating drugs" has been changed to "stimulants or other medications used for symptomatic treatment of fatigue". We agree that this exclusion criterion could potentially cause sample bias, leading to the inclusion of patients with severe, resistant fatigue. This was acknowledged in the discussion section.

Please identify the disease-modifying agent subjects are on. Table 1, please provide disease duration in years.

We identified the disease-modifying agent our study subjects were on. In Table 1, disease duration in years was provided.

Results of regression models to predict fatigue level (low n=14, moderate n=14, high n=15) would be a nice addition, first entering demographics (age, disease duration, EDSS) and T2 lesion volume, and adding structural volumes and DTI indices to the model next.

As requested, we have included a regression analysis. In the model, we included only the variables that were found to have a statistically significant difference between the two groups, so as to avoid overestimating the model. The results are included in Table 2.

My main concern is that (as the authors acknowledge) the sample size is small, multiple comparisons have not been controlled for, and correlational research is limited. The small groups of patients falling into categories of low, moderate, and high fatigue are nicely matched for disease duration, T2 lesion load, and age. However, given the multidimensionality of fatigue (as mentioned by the authors), we must wonder whether relationships between neural level variables and level of fatigue are spurious and/or mediated by a third variable that has not been captured. The sample includes only patients with low disability (EDSS < 4); this should be mentioned in the abstract and the introduction, as it is a strength of the study. This methodological choice should be described in the context of previous MS literature and describe how this may affect results.

We added the information about the low EDSS in the abstract and introduction. We also discussed this methodological choice in the context of previous relevant MS literature in the discussion section.

Language should be tempered in the abstract and discussion to reflect the study limitations. For instance, first sentence of discussion “focus of our study was to establish the structural and neuronal integrity measures of fatigue severity…” With 43 people, nothing has been or will be established.

Agree. Language was modified.

Discussion: “These findings suggest that disruption of the neuronal integrity occurs prior to the evidence of cortical atrophy in the RTC.” This statement, with implications of a temporal sequence for observed (cross sectional) neural-level patterns, is an over interpretation of the study data and should be removed.

This statement was removed, as requested.

The statement “We provide novel evidence about the neuronal integrity measures in non-lesional cortical areas of RRMS patients” was a surprise; no accounting for lesions in Freesurfer or DTI analysis was described in the methods section. Final sentence of discussion the words “striking correlation” may be an overstatement.

Our analysis looked at the DTI measures, namely FA and RD in the cortex of patients with MS. We used Freesurfer to generate cortical masks and evaluated the FA and RD values within the cortical masks. The value of FA and RD in the white matter, where the lesions reside was not included in the analysis, hence, our measures are by definition,non-lesional. We agree that one could argue that a juxtacortical lesion could potentially be a source of error in that it may be misclassified as part of the cortex, however, the method we have presented is essentially non-lesional, since it looks at the DTI measures in the non-lesional areas, namely the gray matter. This statement was modified, as requested.

The "striking correlation" was removed in the final sentence of discussion.

Reviewer 2 Report

Major:

-       “The groups did not differ in age, disease duration, 
medication, EDSS and T2 lesion volume.“ Please provide statistical data for this claim (test statistics and p-values). A quick view on Table 1 suggests that this claim is not correct and that there is at least a significant age difference between the LF and HF groups. If there is an age-association with FSS (as the age difference between the groups suggests), statistical models should be corrected for age.

-       The authors only report positive findings quantitatively. Furthermore, the authors do not correct for multiple testing. In combination this is problematic since it leaves unclear the number of statistical tests performed and the amount of a potential Type I error accumulation. The authors should clearly state how many structures they compared between the groups or investigated an association with to allow better assessment of Type I error accumulation. I also suggest reporting the results of all tests including group means/SDs and test statistics at least in a supplementary table.

-       Please either correct for multiple testing (this can be done additionally and the significance level after correction with e.g. Bonferroni-Holm can be reported next to the uncorrected p-values) or clearly mention non-correction as limitation in the discussion.

Minor:

-       Please provide the freesurfer version

-       Please provide ordinal EDSS scores as Median plus Min/Max or IQR and not as Mean and SD.

-       Please introduce all used abbreviations on their first use in the manuscript (SCP…)

-       Do error bars in figures represent SE or SD? Lease clarify and make clear in figure legends

-       The presentation of DTI results describes correlations simply based on absolute values. While this is correct, the author could aid readers in interpreting the study’s results by qualitatively describing the finding. For example, lower FA is “worse” in the context of this sample. Higher RD is “worse” in the same context. This makes e.g. the use of “In contrast” (p5 lines 139-140) misleading since there is not really a contrast in findings.

Author Response

The authors would like to thank the reviewer for the time he/she invested in improving the quality of our manuscript. The article was accordingly revised.

Major:

-       “The groups did not differ in age, disease duration, 
medication, EDSS and T2 lesion volume.“ Please provide statistical data for this claim (test statistics and p-values). A quick view on Table 1 suggests that this claim is not correct and that there is at least a significant age difference between the LF and HF groups. If there is an age-association with FSS (as the age difference between the groups suggests), statistical models should be corrected for age.

We have provided the statistics and p-values for the variables mentioned above. It should be noted that the age difference between the two groups is well within the SEM, and more importantly, no more than a few years. It is highly implausible that a difference in fatigue level could be attributed to a few years of difference in age (e.g the average 39-year old's vs the average 43-year old's fatigue level). Even if the groups age differences were to be statistically significant (e.g. due to small standard deviation), surely the reviewers would agree that a difference of 4 years is not clinically significant.

In our regression analysis, we have corrected the data for age, as requested by the respected reviewer, however, we performed the regression analysis both with and without age as an explanatory variable. In other words, we have shown what are the morphometric and  diffusion variables that may contribute to the differences observed in the fatigue scale in our group, both with and without correction for age. Although we cannot establish the biological plausibility for the morphological and diffusion findings in our study, given the unlikely clinical significant of the age difference of a few years, we have included both results to allow the reader to choose which is the most appropriate analysis.

-       The authors only report positive findings quantitatively. Furthermore, the authors do not correct for multiple testing. In combination this is problematic since it leaves unclear the number of statistical tests performed and the amount of a potential Type I error accumulation. The authors should clearly state how many structures they compared between the groups or investigated an association with to allow better assessment of Type I error accumulation. I also suggest reporting the results of all tests including group means/SDs and test statistics at least in a supplementary table.

-       Please either correct for multiple testing (this can be done additionally and the significance level fter correction with e.g. Bonferroni-Holm can be reported next to the uncorrected p-values) or clearly mention non-correction as limitation in the discussion.

We have made the requested changes in the paper. We have reported the number of variables included in the analysis, as well as the uncorrected p-values and Bonferroni adjusted p-values. We have also reported the results of all the tests performed, along with the group as mean±S.E.M in a supplementary table.

Minor:

-       Please provide the freesurfer version

Provided, as requested.

∽    Please provide ordinal EDSS scores as Median plus Min/Max or IQR and not as Mean and SD.

Provided, as requested.

-       Please introduce all used abbreviations on their first use in the manuscript (SCP…)

Abbreviations are introduced on their first use in the manuscript.

-       Do error bars in figures represent SE or SD? Lease clarify and make clear in figure legends

Error bars represent SEM. This was clarified in figure legends.

-       The presentation of DTI results describes correlations simply based on absolute values. While this is correct, the author could aid readers in interpreting the study’s results by qualitatively describing the finding. For example, lower FA is “worse” in the context of this sample. Higher RD is “worse” in the same context. This makes e.g. the use of “In contrast” (p5 lines 139-140) misleading since there is not really a contrast in findings.

Agree. Statement was corrected, as requested.

Submission Date

24 June 2017

Round 2

Reviewer 1 Report

The manuscript is now acceptable for publication.

Author Response

Dear Reviewer,

Thank you for your time and constructive comments. We have modified the article as requested.

The difference in age between HF and LF group did not reach statistical significance. In the multivariate regression analysis, age was not a predictor of fatigue severity (borderline p=0.058), but there was a trend between age and fatigue status. This was mentioned in the Results section, under "multivariate regression analysis".

Right temporal lobe fractional anisotropy is the best predictor of fatigue severity among the variables examined with a p-value of 0.011, after correction of age. The p=0.11 was incorrect, was a typo that was corrected. Please see also Table 2b. 

The age variable with the borderline p-value as a predictor of FSS score was also discussed under study limitations study of the Discussion section.

Thank you for accepting our manuscript for publication in the Brain Sciences.